# Effective Establishment of Glide-Path to Reduce Torsional Stress during Nickel-Titanium Rotary Instrumentation

**DOI:** 10.3390/ma12030493

**Published:** 2019-02-05

**Authors:** Ibrahim H. Abu-Tahun, Sang Won Kwak, Jung-Hong Ha, Asgeir Sigurdsson, Mehmet Baybora Kayahan, Hyeon-Cheol Kim

**Affiliations:** 1Department of Conservative Dentistry, School of Dentistry, The University of Jordan, Amman 11942, Jordan; ihtahun@yahoo.com; 2Department of Conservative Dentistry, School of Dentistry, Pusan National University, Dental Research Institute, Yangsan 50612, Korea; endokwak@pusan.ac.kr; 3Department of Conservative Dentistry, School of Dentistry, Kyungpook National University, Daegu 41940, Korea; endoking@knu.ac.kr; 4Department of Endodontics, New York University College of Dentistry, New York, NY 10010, USA; asgeir.sigurdsson@nyu.edu; 5Department of Endodontics, Faculty of Dentistry, Okan University, Istanbul 34947, Turkey; bayborakayahan@yahoo.com

**Keywords:** glide-path, NiTi rotary file, torque measurement, torsional resistance, torsional stress

## Abstract

This study compared the torque generation during canal shaping with a nickel-titanium endodontic instrument according to the extent of glide-path establishment. Seventy-five simulated S-shaped canal blocks were divided into five groups (n = 15) according to the number of repetitive insertions to the working length using a One G glide-path instrument: groups with 5, 10, 15, and 20 insertions as well as group Z without glide-path establishment. When the tip of the One G file reached the working length, the file was moved back and forth repetitively at the working length for the designated number of times for each group. The instrumentation procedure with HyFlex EDM had 15 pecking strokes. During instrumentation, the generated torque was transmitted to a customized data acquisition module and collected using customized software. Data were computed to determine the maximum torque and total stress. The maximum screw-in forces were statistically analyzed using one-way analysis of variance and Tukey’s post hoc comparison test with a significance level of 95%. While the maximum stress did not have significant differences among the five groups (*p* > 0.05), groups with more than 10 repetitive insertions generated lower total stress during instrumentation than did the group with 5 insertions and group Z (*p* < 0.05). Under the limitations of this study, repetitive insertions of glide-path establishment files at the working length reduced stress generation during the shaping using nickel-titanium instruments.

## 1. Introduction

Nickel-titanium (NiTi) rotary instruments allow for well-tapered and clean root canals with a low tendency of aberrations, but they do have some fracture risk during use [1,2,3]. The main mechanisms of file fracture or material failure were revealed as two modes of torsional failure and cyclic fatigue, with the former contributing to a significant proportion of the failures [2]. While cyclic fatigue fracture is caused by repetitive compressive and tensile stresses on the outermost fibers of a file rotating in a curved root canal, torsional failure occurs when the tip of the instrument binds to the canal wall, even in a straight root canal [4]. Establishing a path of least resistance or a “glide-path”, which is defined as a smooth tunnel from the orifice of the canal to the terminus of the root, during the initial preparation has been shown to be a highly critical procedure in preventing early file fracture or failure, mostly due to torsional stress [4,5,6]. The literature reports greater ease for the clinician when either rotary and reciprocating NiTi instruments are employed after a glide-path has been established, possibly giving the clinician more confidence when treating complex and challenging endodontic cases [4,7,8]. 

Stainless steel (SS) K-files (size #10 or #15) were for the longest time the only instruments available for establishing a glide-path [5,6,9,10]. In narrow, tight, and curved canals, a K-file was used with a watch-winding movement in an attempt to remove the obstructing dentin and advance the file to the apex. Once to length, or when an obstruction was encountered, a vertical in-and-out movement was recommended with the amplitude of approximately 1 mm, and then that amplitude was gradually increased until sufficient dentin was shaved away to allow the file to advance to the apex [10]. Once to length again, repeated push-and-pull movements were recommended until the next larger file in the sequence could be inserted easily to the desired working length [11].

However, using SS K-files in this manner is not easy; it requires practice, patience, and time. These issues have become one of the major impetuses for designing NiTi glide-path establishment instruments [7,12]. Now there are many brands of NiTi glide-path establishment instruments available, with tip sizes from #12 to #19 and tapers of 2%, 3%, or progressive changing tapers. These brands include PathFile (Dentsply Sirona, Ballaigues, Switzerland), ProGlider (Dentsply Sirona), G-file (Micro-Mega, Besançon, France), One G (Micro-Mega), and Scout Race (FKG Dentaire, La Chaux-de-Fonds, Switzerland).

Compared to a conventional glide-path using SS hand files, the new rotary technique has been shown to be faster and to maintain the original canal anatomy better, resulting in less distortion of canal curvatures and thereby ultimately leading to fewer canal aberrations [12]. It has been reported that some NiTi glide-path instruments are designed such that they do not cause apical transportation even when the files repeatedly (up to 10 times) reach the apical terminus of the working length (WL) [4,13].

However, so far, there has not been any research reported on whether there is a minimum or an optimal number of repetitive insertions to the working length of these glide-path instruments in an attempt to reduce the torsional stress, i.e., the risk of torsional failure on the canal-shaping NiTi instruments used during the shaping procedure after the glide-path has been established. Therefore, the aim of this study was to compare the torque generation during canal shaping with a NiTi shaping instrument after different numbers of insertions of glide-path files to the full working length.

## 2. Materials and Methods

A One G NiTi file was used to establish a glide-path in a simulated resin canal (Endo Training Block; Dentsply Sirona) with an S-shaped curvature. In these blocks the respective angles and radii of the curvatures were 30° and 5 mm for the coronal curvature, and 20° and 4.5 mm for the apical curvature. The working length, as defined by the tip of a #10 K-file (Dentsply Sirona) becoming visible at the apical foramen, was measured under an operating microscope (Leica M320 F12; Leica Microsystems, Wetzlar, Germany) at 10× magnification.

Seventy-five simulated canal blocks were divided into five groups (n = 15) according to the number of repetitive insertions to the working length: 5, 10, 15, and 20 times of repetition. Group Z was instrumented with the shaping file without glide-path establishment and was assigned as a negative control group.

A One G file was selected in this study to make the glide-path, and it operated with an endodontic motor (X-smart; Dentsply Sirona) in a continuous rotation at 300 rpm and torque set at 1.2 N⋅cm, as recommended by the manufacturer. When the tip of the One G file reached the working length, the file was moved back and forth repetitively at the working length for a designated number of insertions (5, 10, 15, and 20) for each group. During the glide-path establishment procedure, the simulated canal was irrigated with saline using a 30-gauge needle after every five insertions to the working length.

After glide-path establishment, each simulated canal block was enlarged using HyFlex EDM (Coltene/Whaledent, Altstätten, Switzerland), which has a #25 ISO tip and a 8%/variable taper. It was also operated with the same motor at the recommended speed of 500 rpm and torque set at 2.4 N⋅cm in the same continuous rotation motion. The instrumentation procedure was controlled to have the same 15 pecking strokes, and the last three strokes were repeated at the working length. The canal was irrigated with saline every three pecking strokes and the debris on the flutes of the instrument was removed. All of the instrumentation procedures were conducted by one endodontist with 10 years of experience. During instrumentation, the generated torque was extracted from the motor and recorded at a rate of 100 Hz using a customized data acquisition module (Figure 1). Acquired data were analyzed using software (Origin v6.0 Professional, Microcal Software Inc., Northampton, MA, USA) to produce a plot (Figure 2). The maximum torque and total torque (value calculated by integrating the plot of torque changes) were computed by this software.

The data from the five tested groups during the instrumentation procedures were statistically analyzed (SPSS ver. 22; IBM Corp., Somers, NY, USA) using one-way analysis of variance and Tukey’s post hoc comparison tests with a significance level of 95%.

A custom-made test device (DMJ system, Busan, Korea) (Figure 2) was used to measure screw-in forces during file movement. Each resin block containing a pre-enlarged simulated canal and designated file were connected to the device and the file was operated (Figure 2 in box). The rotation speed was set at 350 rpm for all groups and the reciprocating angles were set as 170° counterclockwise and 50° clockwise to give the same kinetic conditions for these reciprocating files. The files were automatically moved to a working length of 16 mm, with a pecking motion with a distance of 4 mm and a crosshead speed set at 1 mm/s. The pecking depth was increased by 1 mm for each pecking motion until the file reached the working length. The screw-in forces generated during the file movement were automatically recorded by using customized software, and the maximum force was extracted from the data.

The maximum screw-in forces during the instrumentation procedures were statistically analyzed (SPSS ver. 22; IBM Corp., Somers, NY, USA) using one-way analysis of variance and Tukey’s post hoc comparison tests with a significance level of 95%.

## 3. Results

The generated total torque and maximum torque values during instrumentation using HyFlex EDM, according to the number of repetitive insertions of the One G glide-path establishment instruments, are presented in Table 1. Representative torque generations from groups 5 and 20 during instrumentation are presented in Figure 2.

While the maximum torque did not exhibit significant difference among the five groups (*p* > 0.05), more than 10 repetitive insertions in groups 10, 15, and 20 generated a lower total torque during instrumentation than did groups 5 and Z (*p* < 0.05). Group Z without the glide-path establishment showed the same total torque generation as groups 5 and 10 (*p* > 0.05). Also, there was no significant different among the groups with 10 to 20 insertion repetitions (*p* > 0.05).

## 4. Discussion

Numerous instrument systems have been designed and manufactured in an attempt to reduce the risk of instrument breakage; however, it has been demonstrated that—irrespective of the file design—the establishment of a good “glide-path” before inserting NiTi rotary shaping instruments significantly reduces the breakage risk [4,5,6]. This is especially true with the recently introduced instruments made of heat-treated alloy (e.g., HyFlex CM (Coltene/Whaledent), ProTaper Gold (Dentsply Sirona), One Curve (Micro-Mega)) as they generally have lower torsional strength than the files made of conventional NiTi instruments [14,15]. Thus, glide-path establishment to reduce the torsional failure is much more important for heat-treated instruments, including the HyFlex EDM file used in this study.

Also, it has been reported that even manual pre-flaring and coronal enlargement using SS files allow for the safer use of NiTi rotary instruments by reducing torsional stresses, and the path also reduces the risk of shaping aberrations [4,5,7,16]. When the canal is narrow, repeated push-and-pull movements using SS files were recommended until the next larger file in the sequence could be placed to the desired working length [11]. Berutti et al. [4] advocated for the glide-path diameter to be at least one size larger than the tip of the first rotary instrument to be used in order to minimize torsional stress on the rotary instrument.

In recent years, NiTi instruments for glide-path establishment have been widely used by clinicians because of the high level of efficiency and convenience that these instruments offer [7,17,18]. However, no specific recommendations, like those for manual SS files, have been published that indicate how to use NiTi glide-path establishment instruments. Ha et al. [13] only reported that 10 repetitive insertions to the working length made a sufficient lumen size for the next shaping instrument without producing apical transportation. However, these studies did not investigate the actual effect that the glide-path establishment could have on the reduction of torsional stress during the movement of the NiTi shaping instruments.

The HyFlex EDM used in this study is a single file system used in a continuous rotation. Other single file systems, such as WaveOne and Reciproc, have the same tip size as the HyFlex EDM for their principal instruments, but work in a reciprocating motion. According to the manufacturers’ recommendations, all of these instruments of single file systems can be used for the entire shaping procedure from orifice to apex. However, if these files are indeed used in this manner, there is an expectation that they will be subjected to significantly higher torsional stresses than instruments of sequential multi-file systems [19]. Therefore, it might even be more important to create a glide-path before using single file systems to provide safer procedural conditions and to lower the risk of torsional failure.

A glide-path of a sufficient size may ensure a reduction in torsional stress, thereby increasing the lifespan of the rotary instrument used for canal preparation [6]. Therefore, this study attempted to evaluate the actual reductions of the torque generated during shaping after various extents of glide-path establishment.

In the present study, standard artificial canals in training resin blocks were used to minimize variations in observation, because of the different size of canal diameter and/or anatomy of the extracted teeth. A preliminary study using extracted teeth demonstrated that even if the extracted teeth were selected after careful radiographic evaluation and all had a similar root curvature, the range of the torque generation had a great and unacceptable variability. On the other hand, under the controlled conditions using a simulated canal, the number of insertion times to the working length showed much better consistency in total torque generation during the instrumentation procedures. The 15 repetitive insertions to the working length significantly reduced the total torque generation, as compared to 5 repetitive insertions at the working length. Also, 10 repetitive insertions had the same effect on the total torque generation as 5 insertions and no glide-path establishment.

Considering this information together with the previous study from Ha et al., 10 repetitive insertions at the working length may give a sufficient size of lumen such that there is a decrease of the torque generation during the shaping procedure [13].

However, all tested groups in the present study had the same level of maximum torque generation during the shaping procedure using the HyFlex EDM instrument. This could be attributed to the anatomical characteristic of the simulated canal used in this study. When the file reached the working length for the first time or passed the maximal curved area, the shaping might have generated the highest torque regardless of the extent of glide-path establishment. Thus, we conclude that glide-path establishment can reduce the total torque generated during shaping, but does not reduce the maximum torque. The results indicate that the risk of file fracture by torsional stress would be reduced by glide-path establishment and preserve the root dentin integrity.

Major advantages of rotary glide-path establishment may include the more thorough initial removal of pulp tissue and debris from the root canal, allowing the operator to maintain working length and patency [9,10,11,20]. An increased flow of irrigation solutions into the apical root canal would be another significant advantage during glide-path preparation by the crown-down procedure. However, clinicians need to consider that the glide-path establishment procedure may also have some shortcomings such as debris extrusion, though less than stainless steel manual instrumentation [13]. This study demonstrated that the amount of debris extrusion and the generated stress during instrumentation was reduced when the canal was instrumented with 3% constant or progressive tapered glide-path files, compared to instrumentation with smaller tapered files such as 2% (e.g., PathFile, Scout Race). It is important in future study to evaluate the changes in the torque generation with different glide-path establishment NiTi files.

The present study has a methodological limitation. Even though the experimental condition was standardized by the use of simulated resin canals, the properties of resin canals are quite different from those of natural teeth. Therefore, further studies need to be conducted on natural teeth, although the varying sizes and shapes of the root canals may result in less reproducible results.

## 5. Conclusions

Under the limitations of this study, it can be concluded that repetitive insertions of glide-path establishment files at the working length may reduce torque generation during the shaping procedures with NiTi instruments. Clinicians should make a serviceable glide-path using sufficient repetitive insertion motions at the working length with a glide-path establishment instrument prior to using NiTi shaping instruments in order to minimize material failure.

## Figures and Tables

**Figure 1 materials-12-00493-f001:**
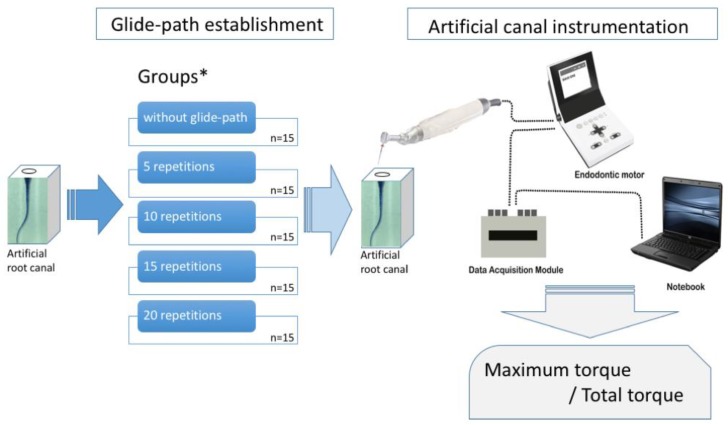
Experimental setting and groups (*) classification according to the number of repetitive insertions of the glide-path instrument to the working length.

**Figure 2 materials-12-00493-f002:**
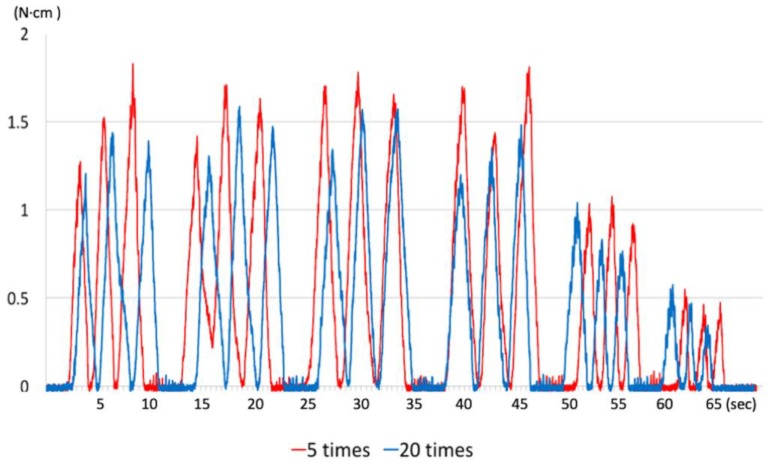
Representative diagrams of torque generations according to the number of repetitive insertions of glide-path instrument at the working length (red plot: group 5, blue plot: group 20). Total torque generations between group 5 (red plot) and group 20 (blue plot) were significantly different (*p* < 0.05; Table 1).

**Table 1 materials-12-00493-t001:** Maximum and total torque (N·cm) generated during instrumentation after repetitive insertions of glide-path instruments at the working length (mean ± SD).

Groups	Z	5	10	15	20
Maximum torque *	1.62 ± 0.55	1.54 ± 0.28	1.56 ± 0.22	1.53 ± 0.28	1.53 ± 0.28
Total torque	30.79 ± 1.24 ^a^	30.97 ± 2.23 ^a^	29.41 ± 2.58 ^ab^	27.75 ± 2.81 ^b^	27.12 ± 1.29 ^b^

*: Maximum torque did not show significant difference among the groups (*p* > 0.05); ^a,b^: different superscripts indicate significant differences among the groups (*p* < 0.05).

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
