# Peer review of "Effective Establishment of Glide-Path to Reduce Torsional Stress during Nickel-Titanium Rotary Instrumentation"

_materials, 2019, doi:10.3390/ma12030493_

Round 1
Reviewer 1 Report
Specific Comments:
At lines 17 and 18, revised wording is recommended. The simulated root canals, not the plastic blocks, are S-shaped.
At line 25, the statement about the data acquisition module enabling computation of the maximum torque and total stress during instrumentation of the root canal does not appear to align with the longer explanation on lines 90 – 94.
At line 79 and subsequently, the authors should replace “Ncm” by “N-cm” or by using the multiplication dot symbol between “N” and “cm”, since the unit of torque is the product of Newton and centimeter.
General comments:
Since this manuscript has been submitted to a materials journal, the text should have a materials science orientation for the background, experimental protocol and findings, and discussion of these results. The Introduction does not contain materials science information about the different types of NiTi alloys that have been used for the rotary instruments as these files evolved during the past two decades. The Discussion does not contain any materials science information that is related to the specific study being reported, such as the mechanisms of instrument wear and the causes of occasional failure, and the research design included no protocols to examine aspects of these important materials science matters. The study contains information that would be of interest to practicing endodontics and to residents and faculty in endodontic programs. The manuscript should be submitted to a journal that focuses on endodontics.
Author Response
Point 1: At lines 17 and 18, revised wording is recommended. The simulated root canals, not the plastic blocks, are S-shaped.
Response 1: We have revised the words’ order.
Point 2: At line 25, the statement about the data acquisition module enabling computation of the maximum torque and total stress during instrumentation of the root canal does not appear to align with the longer explanation on lines 90 – 94.
Response 2: We have revised the sentences for better readability although we could not detail of the methods in the abstract section. The details are well described in the M & M section.
Point 3: At line 79 and subsequently, the authors should replace “Ncm” by “N-cm” or by using the multiplication dot symbol between “N” and “cm”, since the unit of torque is the product of Newton and centimeter.
Response 3: Thank you. All the units were replaced by “N⋅cm”
General comments:
Point 4: Since this manuscript has been submitted to a materials journal, the text should have a materials science orientation for the background, experimental protocol and findings, and discussion of these results. The Introduction does not contain materials science information about the different types of NiTi alloys that have been used for the rotary instruments as these files evolved during the past two decades. The Discussion does not contain any materials science information that is related to the specific study being reported, such as the mechanisms of instrument wear and the causes of occasional failure, and the research design included no protocols to examine aspects of these important materials science matters. The study contains information that would be of interest to practicing endodontics and to residents and faculty in endodontic programs. The manuscript should be submitted to a journal that focuses on endodontics.
Response 4: Thank you for this valuable comments. We have added some sentences regarding the material properties of NiTi alloy and material failures.
Reviewer 2 Report
This paper is a pertinent one, as it explores glide path with new files, so authors add information to this to interesting topic in endodontics.
Nevertheless, some comments in order to improve the manuscript.
Abstract. Add the statistical test used.
Material and methods. Add data about Hyflex EDM (size, taper, one or more files…). Instead of a schematic picture, fig 1 should show the resin block or the data acquisition module, for instance.
What was the motion for each kind of file (One G and Hyflex), rotary, reciprocating? This information must appear in this section, but now this information appears in the Discussion section.
Discussion. First sentence is repeated in the Introduction section. The mentioned previous study should be delete, as we have no information about it.
Author Response
Point 1: This paper is a pertinent one, as it explores glide path with new files, so authors add information to this to interesting topic in endodontics.
Nevertheless, some comments in order to improve the manuscript.
Abstract. Add the statistical test used.
Response 1: The statistical method has been added. Thank you.
Point 2: Material and methods. Add data about Hyflex EDM (size, taper, one or more files…). Instead of a schematic picture, fig 1 should show the resin block or the data acquisition module, for instance.
Response 2: Thank you for this notice. HyFlex EDM is single file system which has #25/.08~variable taper. The size and taper of the tip were added in m & m. The shape of S shaped canal was added to figure 1.
Point 3: What was the motion for each kind of file (One G and Hyflex), rotary, reciprocating? This information must appear in this section, but now this information appears in the Discussion section.
Response 3: Thank you again for this notice. All the instrumentation procedures were conducted in a continuous rotation. The motion for the file systems were added in m & m.
Point 4: Discussion. First sentence is repeated in the Introduction section. The mentioned previous study should be delete, as we have no information about it.
Response 4: Yes, we agree this notice and deleted the repetitive sentences. And we added another sentences to make better plot.
Reviewer 3 Report
Dear Authors,
The aim of this study was to compare the torque generation during canal shaping with NiTi instrument after different number of insertion. The manuscript was well structured and the results would be useful information for dentist. Some minor points should be revised by following comments.
Minor points:
Figure 1
"N=75" should be removed and add "n=15" at the bottom of Groups*.
Page 5, Line 1
"et al" should be used in Italic font. Revise other related points.
Author Response
Dear Authors,
The aim of this study was to compare the torque generation during canal shaping with NiTi instrument after different number of insertion. The manuscript was well structured and the results would be useful information for dentist. Some minor points should be revised by following comments.
Minor points:
Point 1: Figure 1
"N=75" should be removed and add "n=15" at the bottom of Groups*.
Response 1: Thank you for this notice. The figure was re edited.
Point 2: Page 5, Line 1
"et al" should be used in Italic font. Revise other related points.
Response 2: Thank you. It has been revised in the manuscript.
Round 2
Reviewer 1 Report
The revised manuscript has addressed the concerns of this reviewer for the original submission.